# Iterative improvement in the automatic modular design of robot swarms

Jonas Kuckling, Thomas Stützle and Mauro Birattari

IRIDIA, Université Libre de Bruxelles, Brussels, Belgium



## ABSTRACT

Iterative improvement is an optimization technique that finds frequent application in heuristic optimization, but, to the best of our knowledge, has not yet been adopted in the automatic design of control software for robots. In this work, we investigate iterative improvement in the context of the automatic modular design of control software for robot swarms. In particular, we investigate the optimization of two control architectures: finite-state machines and behavior trees. Finite state machines are a common choice for the control architecture in swarm robotics whereas behavior trees have received less attention so far. We compare three different optimization techniques: iterative improvement, Iterated F-race, and a hybridization of Iterated F-race and iterative improvement. For reference, we include in our study also (i) a design method in which behavior trees are optimized via genetic programming and (ii) `EvoStick`, a yardstick implementation of the neuro-evolutionary swarm robotics approach. The results indicate that iterative improvement is a viable optimization algorithm in the automatic modular design of control software for robot swarms.

## INTRODUCTION

In this article, we investigate the use of iterative improvement in the automatic modular design of control software for robot swarms. Swarm robotics is the application of swarm intelligence principles to robotics (*Dorigo, Birattari & Brambilla, 2014*). A swarm of simple robots—simple in terms of hardware and control software—is tasked to solve a mission that none of the individual robots could perform alone. A swarm-level behavior emerges through the interaction each robot has with its neighboring peers and with the environment. By definition, a swarm is highly redundant, self-organized, and decentralized. This offers several advantages such as robustness towards failure of individual robots and scalability of the swarm (*Dorigo, Birattari & Brambilla, 2014*). The main challenge of swarm robotics is to conceive the control software of the individual robots in such a way that a desired collective behavior emerges. A general methodology for the manual design of robot swarms is still missing and existing approaches either operate under restrictive assumptions or are labor-intensive, time-consuming, error-prone, and nonreproducible (*Brambilla et al., 2013*; *Francesca & Birattari, 2016*).

Corresponding author
Mauro Birattari, mbiro@ulb.ac.be

An alternative to manual design is automatic design (*Francesca & Birattari, 2016*). In automatic design, the design problem is reformulated as an optimization problem. Within this optimization problem, solution candidates are instances of control software and an optimization algorithm searches for an optimal candidate with regard to a mission-specific objective function. The optimization process can be run either directly on the robots (on-line design) or in a simulation environment before the generated control software is uploaded to the robots (off-line design) (*Francesca & Birattari, 2016*). In the context of this article, we focus on the automatic off-line design of control software for robot swarms. For an off-line design method to qualify as *automatic*, it must be able to produce control software for a set of distinct missions without the need for any manual intervention (*Birattari et al., 2019*). A popular approach to the automatic off-line design of robot swarms is neuro-evolution (*Trianni, 2008*; *Trianni, 2014*). It applies the general approach of neuro-evolutionary robotics (*Lipson, 2005*; *Floreano, Husbands & Nolfi, 2008*) to swarm robotics: an evolutionary algorithm optimizes an artificial neural network (*Bäck, Fogel & Michalewicz, 1997*). Neuro-evolutionary swarm robotics has been successfully applied to several missions, such as collective movement (*Quinn et al., 2003*), hole avoidance (*Christensen & Dorigo, 2006*), aerial communication relay (*Hauert, Zufferey & Floreano, 2009*), or synchronization of movement (*Trianni & Nolfi, 2009*), yet some limitations remain. The most notable one is the difficulty to reliably cross the reality gap (*Silva et al., 2016*).

An alternative automatic off-line design approach is automatic modular design. Instead of generating a solution from scratch, automatic modular design generates control software by combining and possibly fine-tuning modules that implement low-level behaviors. *Duarte et al. (2016)* proposed an approach that first generates simple modules through the use of an evolutionary algorithm. These modules are then assembled into finite-state machines by a manual designer. *Francesca et al. (2014)* proposed another approach called AutoMoDe. In AutoMoDe, a set of modules, which have been previously designed in a mission-agnostic way and manually implemented, are automatically assembled into instances of control software. AutoMoDe has been successfully applied to assemble modules into finite-state machines (*Francesca et al., 2014*, *2015*; *Hasselmann, Robert & Birattari, 2018*) and behavior trees (*Kuckling et al., 2018a*).

Our main motivation is based on one key observation of *Francesca et al. (2015)*: changing the optimization algorithm from F-race to Iterated F-race resulted in an improvement of performance. The authors, therefore, reasoned, that an important factor for the quality of automatic design approaches is how efficiently the optimization algorithm can make use of the allocated simulation budget. Local search algorithms are a class of optimization algorithms that have shown to efficiently traverse the search space (*Hoos & Stützle, 2005*). Our hypothesis is that local search algorithms might have a place in the automatic design of control software for robot swarms, as they can converge quickly towards promising solutions if the search space is smooth.

Local search algorithms search for an optimal solution with regard to an objective function, by searching through the neighborhood of a current candidate solution. The neighborhood is a set of candidate solutions that are "close" to the current one.

If an improving candidate solution is found in the neighborhood, then it becomes the current one and the local search will continue to explore around it. See the next section for a more formal definition.

We take AutoMoDe-`Chocolate` (*Francesca et al., 2015*) and AutoMoDe-`Maple` (*Kuckling et al., 2018a*), two automatic design methods of the AutoMoDe family, as a starting point for this study. The two methods use the same set of modules (six behaviors and six conditions) and Iterated F-race (*Birattari et al., 2010*) as an optimization algorithm. They only differ in the architecture into which the modules are assembled: finite-state machines and behavior trees, for `Chocolate` and `Maple` respectively. Deriving from these two design methods, we develop multiple design methods, whose optimization algorithm is based on iterative improvement. To better appraise the performance of the design methods based on iterative improvement, we compare them with the base design methods `Chocolate` and `Maple`, `EvoStick` (*Francesca et al., 2015*)—an evolutionary swarm robotics design method—and a design method based on genetic programming, as described by *Jones et al. (2016)*.

*Jones et al. (2016)* designed behavior trees to control a swarm of kilobots in a foraging mission using genetic programming. The results show that the generated behavior trees are well-performing and human-understandable. The main differences between the work of Jones et al. and the studies presented here are the following. They use behavior trees with atomic actions, such as moving forward or turning. These actions never return running for more than one tick consecutively. In contrast, this study uses more high-level behaviors that can run potentially indefinitely. *Jones et al. (2016)* also only consider a single foraging mission, whereas this study aims to compare the performance of automatic design methods over a class of missions.

Our results indicate that iterative improvement is indeed a viable candidate for the automatic modular design of control software. We found no indication that the search space is formed in such a way that it would hinder a local search algorithm to effectively find an optimum.

## NEIGHBORHOOD STRUCTURE FOR ITERATIVE IMPROVEMENT

In this section, we describe the neighborhood that we use for the iterative improvement algorithm. Before doing so, we will give a more formal description of iterative improvement. Iterative improvement is a simple algorithm from the class of local search algorithms, a class of optimization algorithms that operate on the following principles (*Glover & Kochenberger, 2003*). Let $P$ be the optimization problem. $P$ defines a set of solutions, which can take different forms such as variable assignments, ordering of elements, or graph cuts. Often $P$ also defines a set of constraints that restrict the set of solutions. A candidate solution is a solution that satisfies all constraints. Let $C$ denote the set of all candidate solutions. The objective function $f_P : C \to \mathbb{R}$ of $P$ assigns a quality measure to each candidate solution. The goal is to find a solution $c \in C$ that is optimal with respect to $f_P$. This is either a global minimum ($f_P(c) \leq f_P(c')$, $\forall c' \in C$) or a global

maximum ($f_P(c) \geq f_P(c')$, $\forall c' \in C$), depending on the definition of $f_P$. However, any maximization problem can be transformed into a minimization problem (and vice versa) as $\text{argmax}(f_P) = \text{argmin}(f'_P)$ for $f'_P(x) = -f_P(x)$. For the remainder of the paper, we assume that the objective function is to be maximized. For every candidate solution $c$, a neighborhood $N(c)$ is defined. $N(c)$ contains all candidate solutions that can be reached from $c$ through one step of the local search algorithm. Starting from an initial candidate solution, the local search algorithm tries to move from the incumbent candidate solution $c$ to a candidate solution $c' \in N(c)$. The selected candidate solution $c'$ is called the perturbed candidate solution when the neighborhood $N(c)$ is described through the application of perturbation operators. The perturbed candidate solution is accepted—and becomes the new incumbent candidate solution—according to an acceptance criterion. Once a termination criterion is met, the current incumbent candidate solution is returned. The returned solution often is not necessary the global optimum, but only a local optimum ($f_P(c) \geq f_P(c')$, $\forall c' \in N(c)$), that is there is no better candidate solution in the neighborhood of $c$, but it could be that a candidate solution outside of the neighborhood of $c$ has a better solution quality.

Iterative improvement is a local search technique that accepts a perturbed candidate solution if and only if it is better than the current one. There still remains some freedom in implementing this template. For example, the neighborhood could be explored completely before accepting the best solution in the entire neighborhood or the first improvement found in the neighborhood could be accepted without taking the rest of the neighborhood into account. Similarly, the neighborhood can be explored randomly or according to some heuristic. For the rest of this work, we will assume that the neighborhood is explored randomly and the first improvement found is accepted. These assumptions lead to a stochastic hill-climbing search iteratively approaching the first optimum it can find, though not necessarily a global one.

We will define the neighborhood implicitly through a set of perturbation operators. For a given candidate solution (that is, an instance of control software), the neighborhood function can be derived through a systematic and exhaustive application of the perturbation operators.

The perturbation operators follow a hierarchy (structural > modular > parametric) where no lower level should change any properties of a higher one. On the lowest level are parametric perturbations. These perturbations change only the parametric values of the modules (within the limits defined by the reference model), that are associated with the nodes of the instance of control software. They are not allowed to change the modules themselves or alter the structure of the control software. The middle level contains modular perturbations. These perturbations change the modules that are associated with the nodes, but without changing the transitions graph between the modules. They can, however, influence the parametric values of the modules—for example, when replacing one behavior with another, as not all behaviors have the same parameter space. The highest level are structural perturbations that change the graph representation of the control software (either the state transition graph defined by the finite-state machine or the tree

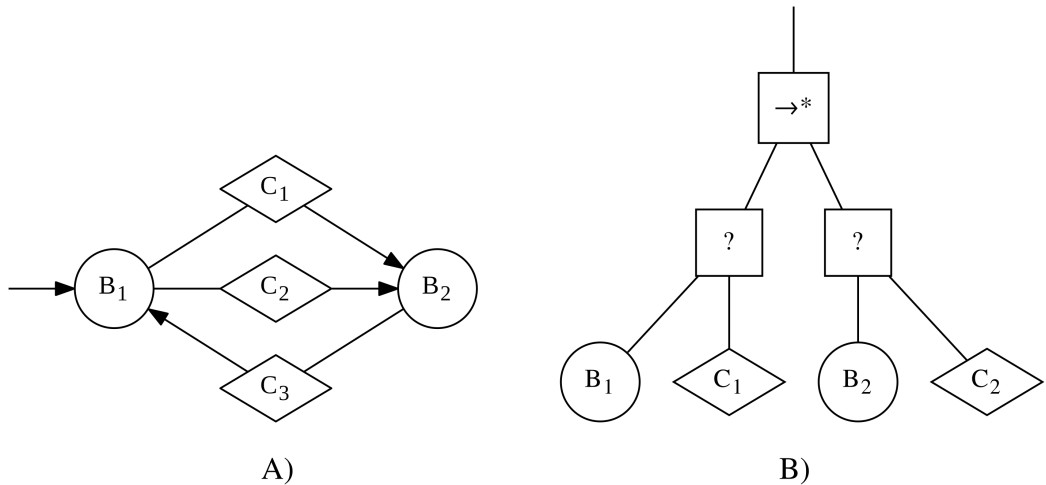

A)                  B)

**Figure 1** **Examples of an instance of control software in the form of a finite-state machine (A) and a behavior tree (B).** The finite-state machine contains two states (represented by circles), with associated behaviors $B_1$ and $B_2$, and a total of three transitions (represented by arrows), with associated conditions $C_1$, $C_2$, and $C_3$ (represented through diamonds).The behavior tree contains a sequence* node ($\rightarrow *$) as the top-level node. Underneath it are two selector nodes (?). Each of the selector nodes has two children: an action node (represented as a circle), with associated behaviors $B_1$ and $B_2$ respectively, and a condition node (represented as a diamond), with associated conditions $C_1$ and $C_2$ respectively.

representation of the behavior trees). These perturbations necessarily influence also both the modular and parametric levels of the control software.

We will present two sets of perturbation operators, which can be used to design finite-state machines and behavior trees, respectively. Nevertheless, the presented iterative improvement algorithm holds for either type of perturbation operator. The operators will be presented in a deterministic fashion and with fixed parameters. Wherever there is a choice, it will either be presented as a fixed parameter or ignored, if not directly relevant to the description of the operator. In any implementation, a choice mechanism needs to be decided upon. In this work, the choices will be made through random and independent sampling from a uniform distribution over all eligible values.

## Finite-state machine

In the context of this work, we will refer to instances of control software as finite-state machines, if they can be formally expressed as probabilistic finite-state machines. Probabilistic finite-state machines are composed of states and transitions. Each state has an associated behavior that is executed as long as the state is active. Transitions connect two states and have an associated condition, which can trigger probabilistically. If the transition triggers, then the current state will become inactive and the state at the other end of the transition will become active and will start to execute its associated behavior. An example of such a finite-state machine is shown in Fig. 1.

In this section, we describe additional properties that a finite-state machine must have to be considered a valid instance of control software. We also describe a set of perturbation operators that transform a valid finite-state machine into another valid one.

*Validity*

In the context of this work, a finite-state machine is considered a valid instance of control software if it fulfills the following criteria:

(V1) Correct number of states

a) **Minimum number of states**: The finite-state machine has at least 1 state.

b) **Maximum number of states**: The finite-state machine has at most 4 states.

c) **Initial state**: Exactly one state of the finite-state machine is designated as the initial state.

(V2) Correct configuration of states

a) **Unique behavior**: Each state has exactly one associated behavior from the set of modules.

b) **Correct parameters**: The parameters of the associated behavior of each state are within the bounds defined for that behavior.

(V3) Correct number of transitions

a) **Minimum number of outgoing transitions**:

1. If there are at least two states in the finite-state machine, then there is no state that has no outgoing transition.

2. If there is exactly one state in the finite-state machine, then this state has no outgoing transition.

b) **Maximum number of outgoing transitions**: Each state has at most 4 outgoing transitions.

c) **Total number of transitions**: There is no limit on the total number of transitions in the graph, other than the implicitly defined $4 \times \#states$.

(V4) Correct configuration of the transitions

a) **Unique condition**: Each transition has exactly one associated condition from the set of modules.

b) **Correct parameters**: The parameters of the associated condition of each transition are within the bounds defined for that condition.

c) **Unique starting point**: Each transition has exactly one starting point.

d) **Unique endpoint**: Each transition has exactly one endpoint.

e) **Not self-referencing**: For each transition the starting and endpoint are different.

*Perturbation operators*

(P1) **Add a new transition to the finite-state machine**: Let $s$ be a state, with less than the maximum outgoing transitions and $s'$ be a different state.
Add a transition from $s$ to $s$.

(P2) **Remove a transition from the finite-state machine**: Let $t$ be a transition from $s$ to $s'$, such that $t$ is neither the only outgoing transition from $s$ nor the only incoming transition into $s'$.
Remove the transition $t$.

(P3) **Add a new state to the finite-state machine**: If the finite-state machine has less than the maximum number of states, let $s'$ and $s'$ be states, where $s'$ has less than the maximum number of outgoing transitions, and $s'$ might be the same state as $s'$.
Add a new state $s$ and add transitions from $s'$ to $s$ and from $s$ to $s'$.

(P4) **Remove a state from the finite-state machine**: Let $s$ be a state that is not the initial state and not an articulation vertex[1] of the state transition graph.
Remove the state $s$ and any incoming and outgoing transitions of $s$.

(P5) **Move the start point of a transition**: Let $t$ be a transition starting in a state $s$, such that $s$ has at least one other transition $t'$ also starting from it. Let $s'$ be a different state than $s$ that has less than the maximum number of outgoing transitions.
Change the start point of the transition $t$ from $s$ to $s'$.

(P6) **Move the endpoint of a transition**: Let $t$ be a transition ending in a state $s$, such that $s$ has at least one other incoming transition $t'$. Let $s'$ be a different state than $s$.
Change the endpoint of the transition $t$ from $s$ to $s'$.

(P7) **Change the initial state**: Let $s$ be the initial state and $s'$ be a different state.
Change the current initial state from $s$ to $s'$.

(P8) **Change condition of a transition**: Let $t$ be a transition.
Change the associated condition of $t$ to a different one from the set of modules.

(P9) **Change behavior of a state**: Let $s$ be a state.
Change the associated behavior of $s$ to a different one from the set of modules.

(P10) **Change parameter of a condition**: Let $t$ be a transition and $p$ a parameter of the associated condition of $t$.
Set a new value for $p$ within the bounds defined for the condition.

(P11) **Change parameter of a behavior**: Let $s$ be a state and $p$ a parameter of the associated behavior of $s$.
Set a new value for $p$ within the bounds defined for the behavior.

The perturbation operators P1–P7 are structural perturbations, as they change the state transition graph, defined by the finite-state machine. The perturbation operators P8 and P9 are modular perturbations as they change the behaviors or conditions associated with the states and transitions of the finite-state machine. The perturbation operators P10 and P11 are parametric perturbations as they only affect the parameters of a single module in the finite-state machine.

## Behavior tree

In the context of this work, we will refer to instances of control software as behavior trees, if they can be formally expressed as probabilistic behavior trees. Behavior trees are trees that are controlled by a "tick", which is a control signal generated with a fixed frequency by the root node. The inner nodes are called control-flow nodes and direct the

[1] In this setting, an articulation vertex is any vertex of the undirected state transition graph, that, if removed, would increase the number of connected components.

tick through the tree. The leaf nodes can be either condition or action nodes and can return one of three values: *success*, *failure* or *running*. Condition nodes observe the environment and return either *success* or *failure*, depending on the state of their associated condition. Action nodes execute a single step of their associated behaviors. They may return *success* or *failure* if the execution leads to a state that can be classified as success or failure of the behavior. Otherwise, they return *running*. Depending on the return value of its child, a control-flow node can either send the tick to another child or return one of the three values itself. For a formal definition, including definitions of the control-flow nodes, see *Colledanchise & Ögren (2018)*. An example of such a behavior tree is shown in Fig. 1.

In order to be able to use the behaviors and conditions of `Chocolate`, a restricted architecture for the behavior trees needed to be put in place (*Kuckling et al., 2018a*). Behavior trees in the restricted architecture have three levels of nodes. The top-level node is a sequence* node. Underneath it are up to four so-called condition-action subtrees; composed of a selector node with a condition node as the first child and an action node as the second child. This architecture allows a behavior in an action node to be executed until the condition in its sibling condition node is met. From the following tick, the next condition-action subtree will receive the tick and execute the next behavior.

In this section, we describe additional properties that a behavior tree must have in order to be considered a valid instance of control software. We also describe a set of perturbation operators that transform a valid behavior tree into another valid one.

### Validity

In the context of this work, a behavior tree is considered to be a valid instance of control software if it fulfills the following criteria:

(V1) **Root node**: The behavior tree has exactly one root node.

(V2) **Top-level node**: The behavior tree has exactly one top-level node (the sole child of the tick-generating root node). The top-level node is a sequence* ($\rightarrow^*$) node.

(V3) **Number of subtrees**: The top-level node has between one and four children, all of which are selector (?) nodes.

(V4) **Condition-action subtree**: Each selector (?) node has exactly two children. The first (left) child is a condition node and the second (right) child is an action node.

### Perturbation operators

(P1) **Add selector subtree**: If the behavior tree has less than the maximum number of selector subtrees, let $i$ be the number of selector subtrees and $0 \leq j \leq i$ be an integer. Add a selector subtree after the $j$th child of the top-level node ($j = 0$ meaning the new tree is added as the first child).

(P2) **Remove selector subtree**: If the behavior tree has at least two selector subtrees, let $t$ be a selector subtree.
Remove the selector subtree $t$ from the tree.

(P3) **Change subtree order**: If the behavior tree has at least two selector subtrees, let $t$ be a selector subtree, $i$ be the number of selector subtrees, $j$ the position of $t$ in the tree, and $1 \leq k \leq i$ be an integer, but $k \neq j$.
Move the selector subtree $t$ to be the $k$th child of the top-level node.

(P4) **Change condition of condition node**: Let $c$ be a condition node.
Change the associated condition of $c$ to a different one from the set of modules.

(P5) **Change behavior of action node**: Let $a$ be an action node.
Change the associated behavior of $a$ to a different behavior from the set of modules.

(P6) **Change parameter of a condition**: Let $c$ be a condition node and $p$ a parameter of the associated condition of $c$.
Set a new value for $p$ within the bounds defined for the condition.

(P7) **Change parameter of a behavior**: Let $a$ be an action node and $p$ a parameter of the associated behavior of $a$.
Set a new value for $p$ within the bounds defined for the behavior.

Perturbation operators P1–P3 are structural perturbations, that is, they change the graph representation of the behavior tree.In our specific restricted architecture, P3 has no structural effect on the graph structure, as the condition-action subtrees are structurally identically, reordering them does not change the overall structure of the behavior tree. However, it has the potential to create structural changes in a less restricted setting, therefore it is justified to classify it as a structural perturbation as well. P4–P5 are modular perturbations and P6–P7 are parametric perturbations, as they only influence a single module or its parameters, respectively.

# AUTOMATIC DESIGN METHODS

We compare ten different design methods: Chocolate, Maple, Minimal-FSM, Minimal-BT, Random-FSM, Random-BT, Hybrid-FSM, Hybrid-BT, GP, and EvoStick. All ten design methods are based on the reference model RM1.1 (*Hasselmann et al., 2018*) for the e-puck robot (see Table 1). The reference model provides access to four types of sensors: proximity, light, ground, and a range-and-bearing board. Additionally, the velocity of the left and right wheel of the e-puck can be adjusted independently. The control cycle period is 100 ms, that is every 100 ms the readings of the sensors are updated and the control software is invoked.

All design methods have the same set of six modules and six conditions available. The six behavioral modules are STOP, EXPLORATION, PHOTOTAXIS, ANTI-PHOTOTAXIS, ATTRACTION, REPULSION. The six conditional modules are FIXED PROBABILITY, NEIGHBORHOOD COUNT, INVERTED NEIGHBORHOOD COUNT, GREY FLOOR, BLACK FLOOR, WHITE FLOOR. For formal definitions of all the modules, see *Francesca et al. (2014)*. In the following, we will describe the individual design methods in greater detail. Implementations of all design methods are available as a download from the Supplemental Materials page (*Kuckling, Stützle & Birattari, 2020*).

## Chocolate and Maple

Chocolate (*Francesca et al., 2015*) and Maple (*Kuckling et al., 2018a*) are two methods that automatically design control software for swarms of e-puck robots. They operate on the same set of modules and adopt the same optimization algorithm; they differ only in

**Table 1 Reference model RM1.1 (*Hasselmann et al., 2018*). Sensors and actuators of the e-puck robot. The period of the control cycle is 100 ms.**

| Sensor/actuator | Parameters | Values |
|---|---|---|
| Proximity | $prox_i$, with $i \in \{0, \ldots, 7\}$ | $[0,1]$ |
| Light | $light_i$, with $i \in \{0, \ldots, 7\}$ | $[0,1]$ |
| Ground | $ground_i$, with $i \in \{0, \ldots, 2\}$ | {*black, gray, white*} |
| Range-and-bearing | $n$ | $i \in \{0, \ldots, 19\}$ |
| | $V_d$ | $([0,0.7]m, [0,2\pi]$ radian) |
| Wheels | $v_l, v_r$ | $[-0.12, 0.12]$ m/s |

the control architecture into which the modules are assembled. `Chocolate` generates finite-state machines with one to four states, where each state has up to four outgoing transitions. `Maple` assembles the modules into behavior trees that have a sequence* node as the top-level node and between one and four selector subtrees, consisting of a selector node with a condition and action node as only children.

Both methods use Iterated F-race (*Balaprakash, Birattari & Stützle, 2007*; *López-Ibáñez et al., 2016*) as the optimization algorithm. Iterated F-race executes several iterations, each of them is reminiscent of a race. In each iteration, a set of candidate solutions is generated and sequentially evaluated on different instances. If a solution performs significantly worse than at least another one (determined by a Friedman test), it is eliminated and will not be evaluated on the remaining instances, effectively allowing computational resources to be used for more promising solutions. The set of candidate solutions for the following iteration is created by sampling around the candidate solutions that have not been eliminated in the previous iterations.

## Minimal-FSM and Minimal-BT

`Minimal-FSM` and `Minimal-BT` are two design methods based on iterative improvement. Algorithm 1 shows a template for the iterative improvement algorithm. The iterative improvement algorithm starts from an initial candidate solution, in our case an instance of control software, and randomly applies one perturbation. If the perturbation leads to an improved candidate solution, the resulting candidate solution is accepted and becomes the new incumbent candidate solution and thus the base for the next perturbation. Otherwise, the result of the perturbation is discarded and a new random perturbation is applied to the candidate solution. This process is repeated until a termination criterion is met. An implementation of the template would need to define three features of the template: the termination criterion, the acceptance criterion, and a neighborhood function.

The function `TerminationCriterion` is used to define an end for the potentially time-consuming search for improvements. Depending on the problem context, this could, for example, be a known optimum value, a criterion that models the passage of time (e.g., runtime of the algorithm, budget of perturbations), or a criterion that detects stagnation for example, indicating a local optimum has been reached.

The acceptance criterion `Acceptance` defines under which circumstances the perturbed solution is selected as the new best solution. In problems with deterministic

---

**Algorithm 1** Iterative Improvement.

**input** : $s_0$ - initial solution
**output** : $s_{best}$ - best encountered solution

$s_{best} \leftarrow s_0$
**while not** `TerminationCriterion` **do**
  $s_{perturbed} \leftarrow$ `RandomNeighbor`$(s_{best})$
  **if** `Acceptance`$(s_{best}, s_{perturbed})$ **then**
    $s_{best} \leftarrow s_{perturbed}$
  **end**
**end**
**return** $s_{best}$

---

performance measures, this might be as simple as comparing the performances of the two solutions. If the performance measure is stochastic, the acceptance criterion could require comparing the mean or median of a sample of performances, or selecting randomly, instead of deterministically, with regard to the observed performance.

The neighborhood function is a problem specific function that takes one solution candidate as input and returns a set of candidate solutions that are considered for the next improvement step. The function `RandomNeighbor` returns a randomly selected element of the neighbor set. This is necessary so that not always the same perturbations are chosen. The neighborhood itself is often defined implicitly through the application of perturbation operators.

Both `Minimal-FSM` and `Minimal-BT` implement this template with one of the neighborhoods defined in the previous section, a termination criterion that counts the number of executed simulations, and an acceptance criterion that compares the means of two samples of executions. The two instances of control software are evaluated in simulation on ten different instances of the mission (initial positions and headings of the robots, defined through a random seed). To avoid overfitting to these exact ten seeds, every time a perturbed controller needs to be evaluated, the oldest two seeds are discarded and replaced by two new ones. The perturbed controller is then evaluated on all ten seeds, while the current best controller is evaluated only on the two new seeds, as it has been already evaluated on the old seeds.

`Minimal-FSM` and `Minimal-BT` take a minimal instance of control software as the initial solution. The minimal instance of control software is an encoding of the stop behavior. In the case of Minimal-FSM, this is a finite-state machine with exactly one state and no transitions. The state has the associated behavior module STOP. `Minimal-BT` is initialized with a minimal behavior tree, that contains the sequence* node as the top-level node. Underneath it is a single selector node with a FIXED PROBABILITY condition node and a STOP action node.

## Random-FSM and Random-BT

`Random-FSM` and `Random-BT` use the same optimization algorithm as `Minimal-FSM` and `Minimal-BT`. The only difference between the respective pairs of design methods is the initial solution. `Random-FSM` and `Random-BT` use a randomly generated valid instance of control software as the initial solution.

### Hybrid-FSM and Hybrid-BT

Hybrid-FSM and Hybrid-BT are hybridizations of Chocolate and Maple and the local search approach already defined for Minimal and Random. They operate in two stages, each stage running for one half of the allocated total budget. In the first stage, the hybrid algorithm creates a candidate solution following the protocol described for Chocolate and Maple. The resulting instance of control software is then supplied to the iterative improvement algorithm under the same setup as for Minimal or Random and is then improved for the remaining budget.

### GP

For comparison, we implement a genetic programming approach, as described by *Jones et al. (2016)*, and apply it to our restricted behavior tree topology. As *Jones et al. (2016)*, we use the DEAP library (*Fortin et al., 2012*) to implement the algorithm and we have chosen the same parameters for the genetic programming. We have adapted the generation count (implicitly defining the available budget) to match the budget used for the other design processes. Additionally, some adjustments had to be made to the mutation operators, adapting them to work with our restricted architecture.

The operator *uniform mutation* was subject to a few implementation changes but the actual effect remains untouched. A random node of the behavior tree is chosen and the subtree defined by this node is replaced with a newly generated subtree.

The *shrink* operator needed a new definition. In the original work, this operator would select a random node and replace it with one of its subtrees. This would change two properties. First, the depth in this subtree would shrink, and secondly, the number of actions would, most likely, be reduced too. Due to the fixed architecture, we cannot incorporate this change, as a child is never a valid replacement for the parents. Instead, one random selector-subtree is pruned from the behavior tree. This has a similar effect of reducing the number of actions in the tree.

In the original definition of the *node replacement* mutation, a random node would be selected and replaced by another node that takes the same number and type of children. In the restricted architecture, this again is not possible, since the type of all inner nodes is fixed. Instead, the node replacement will choose a random leaf node (either an action or condition node) and replace it with another action or condition node. Action nodes are always replaced by another action node and condition nodes always by another condition node. This mimics the effect of the original operator when applied to the leaf nodes while preserving the validity of the instance of control software.

The *ephemeral constant mutation* could be kept completely unchanged and will replace one ephemeral constant value (in the form of a parameter of either the action nodes or the condition nodes) by another in the valid range.

### EvoStick

EvoStick (*Francesca et al., 2014*, *2015*) uses an evolutionary algorithm to optimize the weights of an artificial neural network with a fixed topology. The artificial neural network contains 24 input and 2 output nodes that are mapped to the possible inputs and outputs

as defined by the reference model RM1.1. It is fully connected, feed-forward, and does not contain hidden layers.

## EXPERIMENTAL SETUP

In this section, we describe the experimental protocol used to obtain the results that we will report in the following section.

### Mission

Each design method was used to produce control software for four missions: AGGREGATION WITH AMBIENT CUES (AAC), SHELTER WITH CONSTRAINT ACCESS (SCA), FORAGING, and GUIDED SHELTER. All missions were conducted in a dodecagonal arena with walls of 0.66 m length, enclosing an area of 4.91 m$^2$. At the beginning of each experimental run, the robots are distributed randomly in this arena and the duration of each experimental run is 120 s.

### AAC

In AGGREGATION WITH AMBIENT CUES (AAC), the robots have to aggregate on a black spot. They have two other ambient cues available to help them find the black spot: a light source next to the black spot and a white spot (see Fig. 2).

The black target region is a circular area with a radius of 0.3 m. It is located in the front half of the arena, 0.6 m away from the center of the arena. A light source is placed in front of the black spot (outside of the arena). The arena also contains a white circular area with a radius of 0.3 m. It is located in the back half of the arena, 0.6 m away from the center of the arena.

The objective function is defined as:

$$F_{AAC} = \sum_t \#\text{robots on the black spot} \qquad (1)$$

It computes the cumulative time that robots spend on the black spot. In order to maximize the objective function, the robots have to aggregate as soon as possible on the black spot.

### SCA

In SHELTER WITH CONSTRAINT ACCESS (SCA) the robots have to aggregate inside a shelter. The arena contains the shelter, a light source, and two black spots (see Fig. 2).

The shelter is a white rectangular area, of 0.15 m × 0.6 m, bordered by walls on three sides, and only open on one side. The shelter is positioned in the middle of the arena and is open towards the front of the arena. In the front, outside of the arena, is a light source. Two black circular areas with a radius of 0.3 m are located to the left and right side of the shelter, 0.35 m away from the edge of the shelter.

The objective function is defined as:

$$F_{SCA} = \sum_t \#\text{robots in the shelter area} \qquad (2)$$

It computes the cumulative time that robots spend in the shelter. In order to maximize the objective function, the robots, therefore, have to move into the shelter as soon as possible.

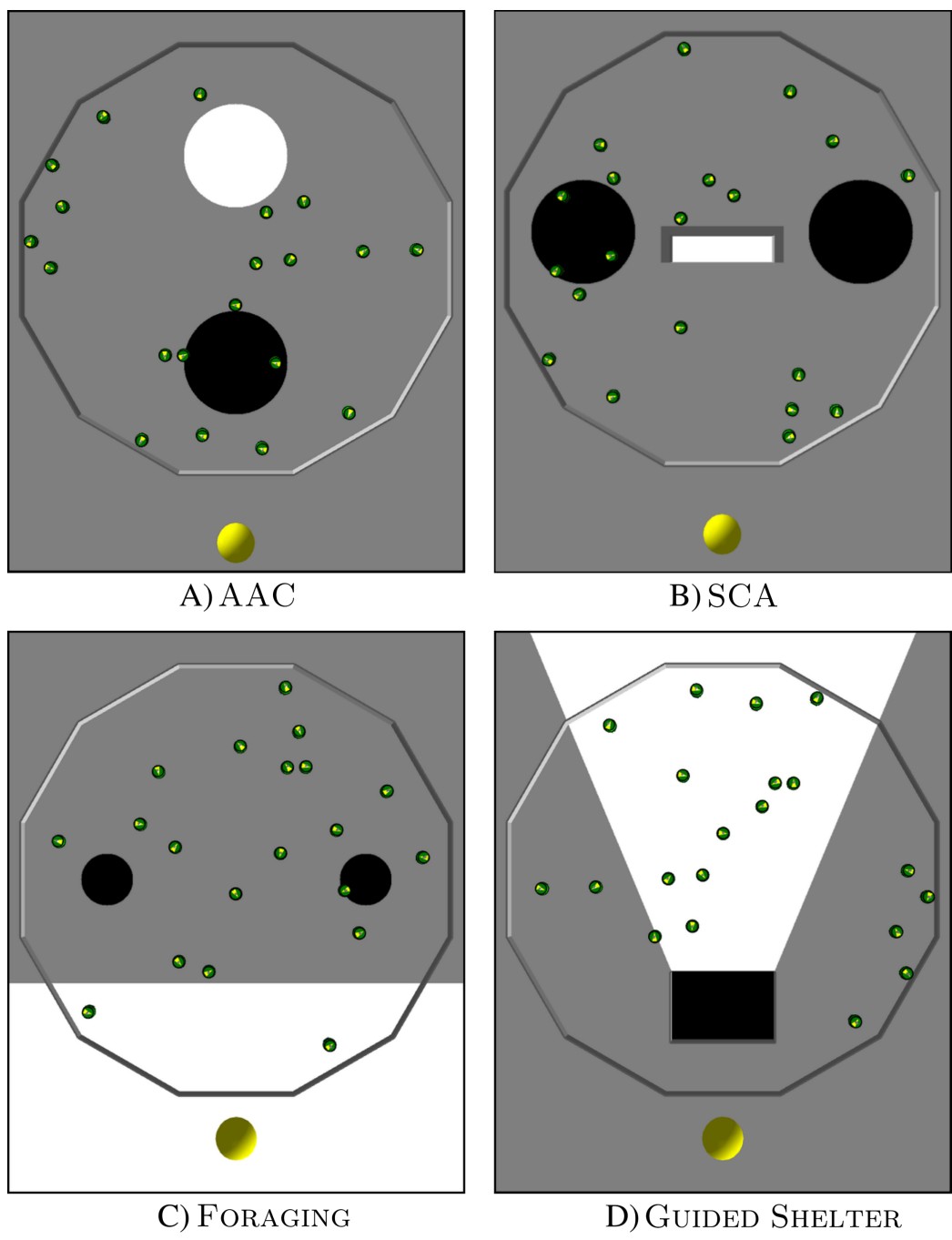

**Figure 2 Arena layouts for the four missions.** The images show ARGoS3 simulations with 20 e-pucks inside a dodecagonal arena with different floor colors. A light source is located at the lower end of each image. (A) AAC, (B) SCA, (C) FORAGING and (D) GUIDED SHELTER.

## Foraging

FORAGING represents an abstracted foraging task. In this task, the robots have to transport items from two food sources to the nest (see Fig. 2). Since an e-puck does not have the capabilities to physically pick up or deposit items, these actions are abstracted to happen when a robot passes over the corresponding area.

A white region in the front side of the arena represents the nest. It covers the whole width of the arena and has a depth of 0.63 m. A light source is located in front of the white region (outside of the arena). Two black circles, with a radius of 0.15 m, represent the food sources. They are positioned 0.45 m away from the nest area and with a distance of 1.2 m between them.

The objective function is defined as:

$$F_{\text{For}} = \#\text{retrieved items} \qquad (3)$$

It computes the number of items retrieved by the swarm. In order to maximize the objective function, the robots have to move back and forth between the nest and a food source as many times as possible.

## Guided Shelter

In GUIDED SHELTER, the robots have to move into a shelter. They have two ambient cues to guide them towards the shelter: a light source and a conic region providing a path to the shelter (see Fig. 2).

The shelter is a black rectangular area of size 0.4 m × 0.6 m. The shelter is located in the front part of the arena 0.3 m away from the front wall of the arena. It is enclosed with walls on three sides and only open towards the far end of the arena. In front of the shelter (and outside of the arena) is a light source. Behind the shelter is a white conic area, that is defined in such a way that the edges of the conic area go through the front corners of the shelter and meet in the center of the light source.

The objective function is defined as:

$$F_{\text{GS}} = \sum_{t} \#\text{robots in the shelter area} \qquad (4)$$

It computes the cumulative time that robots spend within the shelter. In order to maximize the objective function, the robots have to move into the shelter as soon as possible.

## Protocol

The design methods presented in the previous section are used to automatically produce control software for a swarm of 20 e-puck robots. As the design process is stochastic, we repeat it 10 times for every design method, leading to 10 instances of control software per design method. Each of these 10 instances of control software is then assessed 10 times in the design context and 10 times in a *pseudo-reality* context (*Ligot & Birattari, 2018*) to obtain the performance results. The pseudo-reality context is, like the design context, a realistic simulation of the control software. It uses however a different model of reality (i.e., different noise settings). The changed model is sufficient to recreate a performance disparity of the same instance of control software when assessed in the design context and the pseudo-reality context. Prior research has shown that the magnitude of this performance disparity is an indicator of the ability to cross the reality gap reliably (*Ligot & Birattari, 2018*, *2019*). Simulations are performed with ARGoS3, version beta 48 (*Pinciroli et al., 2012*; *Garattoni et al., 2015*). The noise models for the design context and the pseudo-reality context are shown in Table 2. The noise for the proximity,

**Table 2 Noise models for the design and pseudo-reality context.**

| sensor/actuator | Design model | Pseudo-reality model |
|---|---|---|
| Proximity sensor | 0.05 | 0.05 |
| Light sensor | 0.05 | 0.90 |
| Ground sensor | 0.05 | 0.05 |
| Range-and-bearing board | 0.85 | 0.90 |
| Wheels | 0.05 | 0.15 |

light and ground sensors is sampled from a uniform distribution over the interval $[-p, p]$, where $p$ is the noise value reported in Table 2. For the range-and-bearing board, $p$ denotes the probability with which a message lost occurs and for the wheels the noise will be sampled from a gaussian distribution with mean 0 and standard variance $p$.

All design methods are tested with design budgets of 12,500 (12.5 k), 25,000 (25 k), and 50,000 (50 k) simulation runs. That is, after exhausting the allocated simulations the design method must have returned their final instance of control software.

The source code for the experiments, the generated instances of control software, and the details of the performance are available online as Supplemental Material (*Kuckling, Stützle & Birattari, 2020*).

# RESULTS

In the following section, we will make statements such as "method A outperforms method B" or "method A performs significantly better than method B". These statements are based on a paired two-sided Wilcoxon signed-rank tests with a confidence level of 95%. The results of all statistical tests performed can be found in the Supplemental Material (*Kuckling, Stützle & Birattari, 2020*).

## Results 12.5 k

Figure 3 shows the performance of all design methods when they are allocated a budget of 12,500 (12.5 k) simulation runs.

In all missions except GUIDED SHELTER, the design methods `Maple`, `Chocolate`, `Minimal-BT`, `Minimal-FSM`, `Random-BT`, `Random-FSM`, `Hybrid-BT`, `Hybrid-FSM` perform similar to their counterpart that uses the alternative architecture (behavior tree or finite-state machine). The only exceptions are `Hybrid-BT` and `Hybrid-FSM`. In the mission AAC, `Hybrid-FSM` is significantly better than `Hybrid-BT`, and in the mission FORAGING, `Hybrid-BT` is significantly better than `Hybrid-FSM`. For the design methods generating control software in the form of finite-state machines, the performance we registered for `Minimal-FSM`, `Random-FSM`, and `Hybrid-FSM` is similar, with few exceptions. In the mission AAC, `Hybrid-FSM` outperforms `Chocolate`, `Minimal-FSM`. In the missions FORAGING and SCA, `Chocolate` is outperformed by `Minimal-FSM` and `Random-FSM`. As for the design methods generating control software in the form of behavior trees, the performance we registered for all design methods is similar, except for `Hybrid-BT` outperforming AutoMoDe-`Maple` in the mission GUIDED SHELTER.

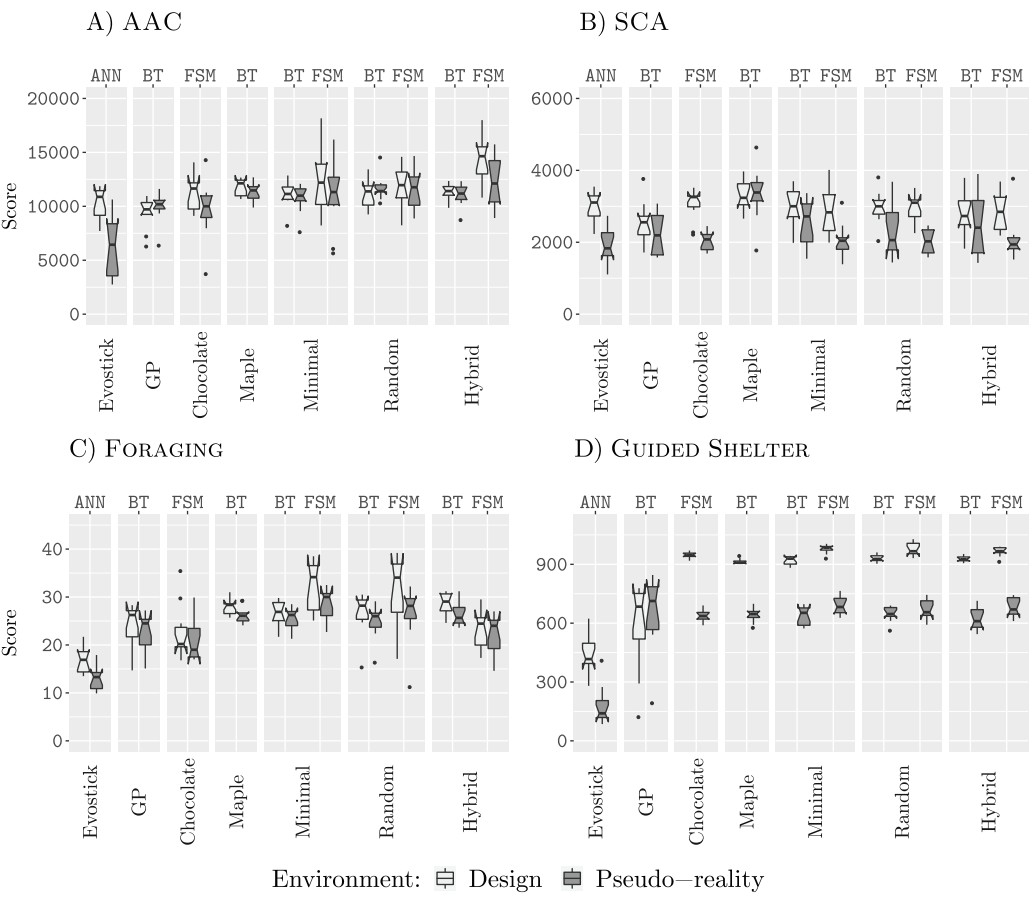

**Figure 3 Results for all design methods for a budget of 12.5 k.** (A) AAC, (B) SCA, (C) FORAGING and (D) GUIDED SHELTER.               

Some of these differences in performance can be explained when looking at the generated control software. For example, in the mission GUIDED SHELTER, where `Chocolate` is outperformed by `Minimal-FSM` and `Random-FSM`. Both `Minimal-FSM` and `Random-FSM` are generating finite-state machines that work after the following principle: A combination of EXPLORATION and ANTI-PHOTOTAXIS steers the robots onto the white area. Once the robot is on the white area, it will use PHOTOTAXIS to move towards the shelter. If the robot leaves the white area and enters the grey area again, it will fall back to the combination of EXPLORATION and ANTI-PHOTOTAXIS. The details of this strategy can vary from instance to instance, for example, the best instance of control software generated by `Minimal-FSM` (see the Supplemental Material (*Kuckling, Stützle & Birattari, 2020*)) starts with the PHOTOTAXIS behavior. For `Chocolate`, on the other hand, the instances of control software only make use of either ANTI-PHOTOTAXIS or Exploration but never both. Visual inspection of the resulting behavior shows, that while instances of control software generated by `Chocolate` still solve the task sufficiently, the robots join the white area on average further away from the nest than instances of control software that are making use of a combination of ANTI-PHOTOTAXIS and EXPLORATION. Thus the robots take longer to join the nest and therefore achieve lower scores on the objective function. A similar

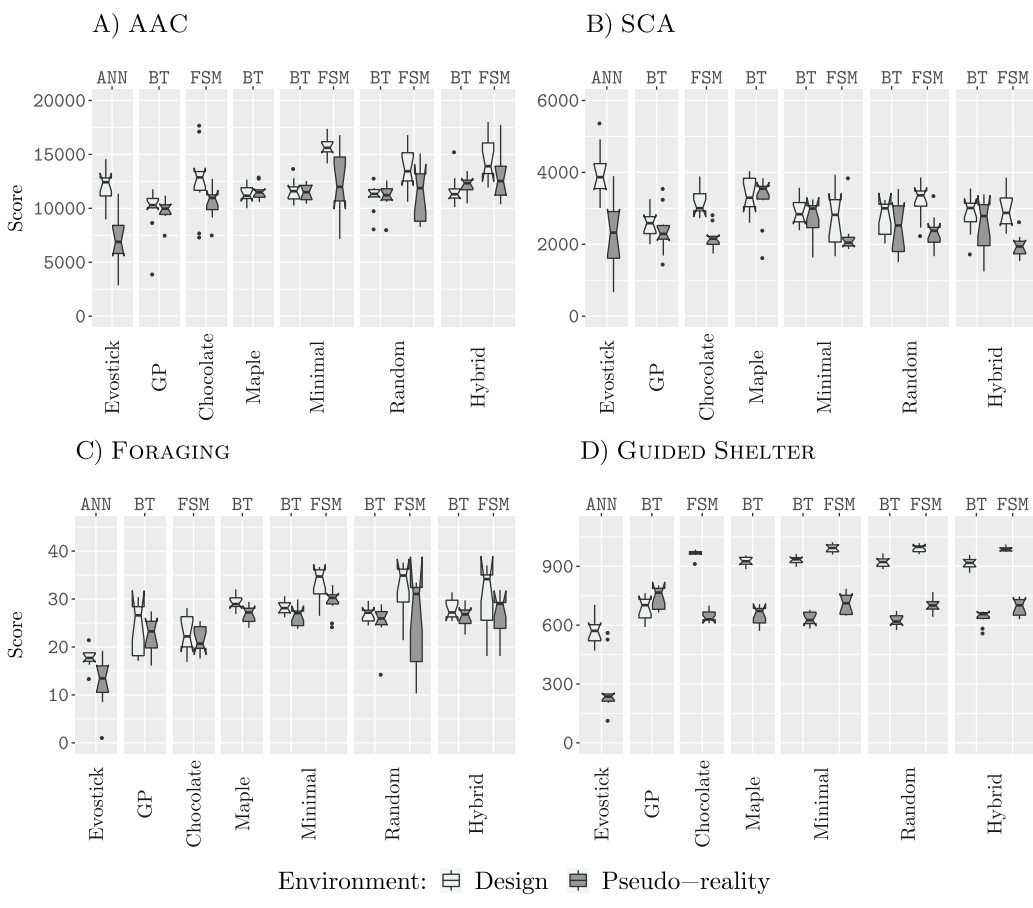

**Figure 4 Results for all design methods for a budget of 25 k.** (A) AAC, (B) SCA, (C) FORAGING and (D) GUIDED SHELTER.

argument can be made for the difference in performance between the generated finite-state machines and the generated behavior trees. In fact, the restricted behavior trees cannot express the same forth-and-back between ANTI-PHOTOTAXIS and EXPLORATION as in those instances generated by `Minimal-FSM`.

In all four missions, `EvoStick` exhibits a large drop in performance, when assessed in pseudo-reality. The other design methods do not exhibit such a large drop, except in the mission SCA, where every design method except for `GP`, `Maple` and `Hybrid-BT` suffers from a large performance drop when assessed in pseudo-reality. This is an indicator that these design methods have a good transferability, allowing them to cross the reality gap satisfactorily.

## Results 25 k

Figure 4 shows the results for all design runs with a budget of 25,000 (25 k) simulations.

In three of the four missions (AAC, FORAGING, and GUIDED SHELTER), `Minimal-FSM` and `Random-FSM` outperform their respective counterparts `Minimal-BT` and `Random-BT`. Additionally, `Hybrid-FSM` outperforms `Hybrid-BT` in the missions AAC and GUIDED SHELTER. In missions AAC, FORAGING, and GUIDED SHELTER, the design methods `Minimal-`

FSM and Hybrid-FSM are able to outperform Chocolate. Additionally, Random-FSM outperforms Chocolate in the missions FORAGING and GUIDED SHELTER. For the design methods Maple, Minimal-BT, Random-BT and Hybrid-BT we register similar performance throughout all four missions, although in the mission FORAGING Random-BT outperforms Maple and Minimal-BT. EvoStick is able to generate sufficiently good control software in the design context, outperforming several other methods.

Comparison of the generated behavior trees in the mission FORAGING yielded no apparent different strategies, and as such the differences in performance can only be explained in such a way that Random-BT was able to select more effective parameters than Maple or Minimal-BT. Chocolate on the other hand fails to make use of many strategies discovered by the iterative improvement based design methods. In the mission FORAGING, Minimal-FSM, Random-FSM, and Hybrid-FSM make use of ANTI-PHOTOTAXIS to navigate out of the nest, after dropping off an item. This strategy provides a two-fold benefit. First, it allows the robots to escape the nest area quicker as if EXPLORATION was used. In the latter case, the robots would need to encounter first and obstacle (either the wall of the arena or another robot), before they could and face towards the rest of the arena again. Secondly, as the robots have been using PHOTOTAXIS to navigate from the sources to the nest, ANTI-PHOTOTAXIS will turn the robots in such a way that they are facing towards the source again. While the embedded obstacle avoidance and interference from other robots will reduce the chance of directly going back to the source, this exploitation of the direction seems to provide a benefit to the performance. Similarly, Chocolate also fails to discover the aforementioned ANTI-PHOTOTAXIS-EXPLORATION strategy in the mission GUIDED SHELTER. In the mission AAC, Chocolate employs the same strategies as the other design methods, although it seems to fail at selecting the appropriate parameters.

When assessed in a pseudo-reality context, however, EvoStick is the worst performing design method, being outperformed by all other methods in the three missions AAC, FORAGING, and GUIDED SHELTER. In SCA, EvoStick performs best in the design context and although it suffers from a significant drop of performance when assessed in pseudo-reality, it still ranges as one of the best performing design methods in pseudo-reality.

In the mission SCA, the designed finite-state machines suffer from a larger performance drop when assessed in pseudo-reality. This could be an indicator that these instances of control software are overdesigned for the specific design context and will not transfer well into reality. Yet, the performance in pseudo-reality is still similar to the performance of the generated behavior trees in pseudo-reality.

## Results 50 k

Figure 5 shows the results of all design methods for a budget of 50,000 (50 k) simulations. In three of the four missions (AAC, FORAGING, and GUIDED SHELTER), the best design method for finite-state machines is able to outperform the best design method for behavior trees. This further supports our previous understanding that behavior trees, in our restricted topology, are too constrained and do not provide the same expressiveness as the finite-state machines (*Kuckling et al., 2018a*, *2018b*). In the missions AAC, SCA and FORAGING, we register lower performance for those behavior trees designed by GP than

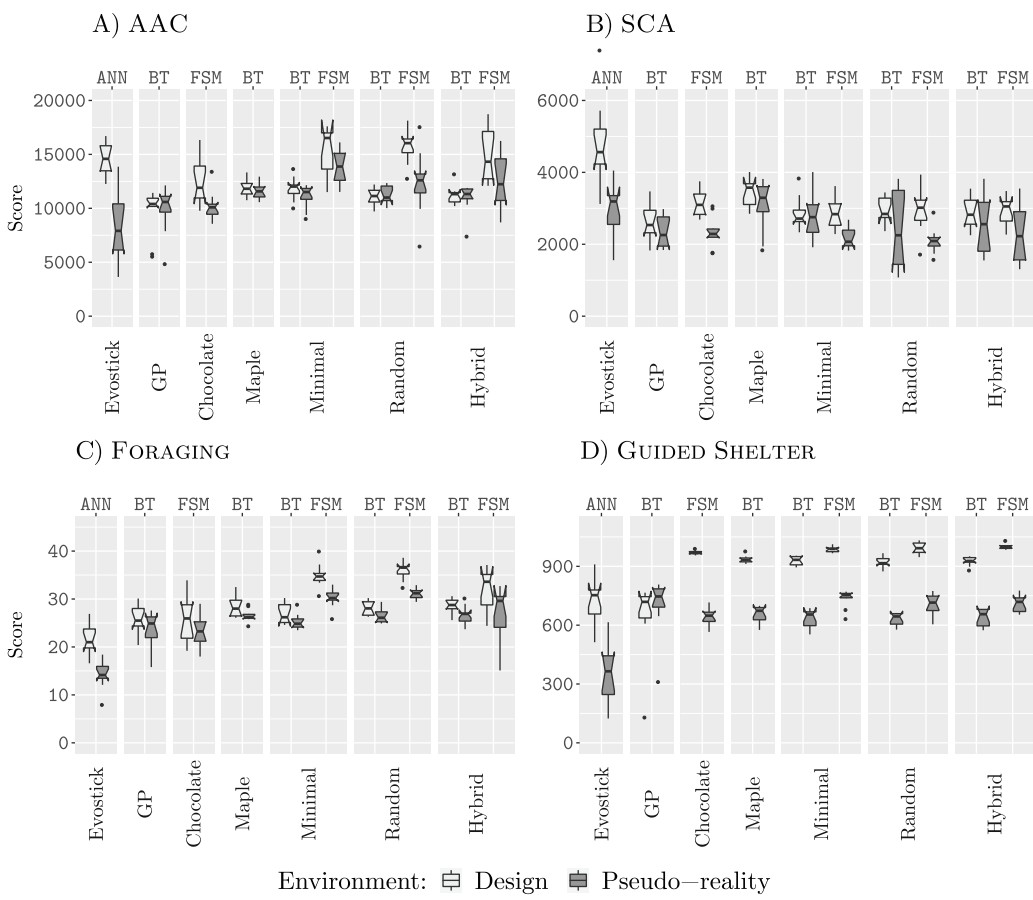

**Figure 5 Results for all design methods for a budget of 50 k.** (A) AAC, (B) SCA, (C) FORAGING and (D) GUIDED SHELTER.

those designed by the other methods. All three local search approaches, `Minimal`, `Random`, and `Hybrid`, perform similarly through all four missions (for a fixed architecture). Finite-state machines designed by those design methods based on iterative improvement outperform those finite-state machines that are designed by `Chocolate`, in the missions AAC, FORAGING, and GUIDED SHELTER.

As for the design budget of 25,000 simulation runs, `Chocolate` fails to discover some of the strategies employed by the other design methods. Most notably, `Chocolate` is neither exploiting the ANTI-PHOTOTAXIS module in FORAGING nor being able to discover the ANTI-PHOTOTAXIS-EXPLORATION strategy to discover the white area in GUIDED SHELTER more quickly.

Throughout all four missions, `EvoStick` suffers from the largest pseudo-reality gap. The control software generated in the form of behavior trees shows only small drops in performance when assessed in pseudo-reality, as in the previous experiments. The finite-state machines designed by `Chocolate` also experience small drops of performance, while the finite-state machines generated by iterative improvement, show larger drops of performance when assessed in pseudo-reality. This can be an indicator of potential overdesign for these design methods, however, the effect is not as big as for `EvoStick`.

## DISCUSSION

We did not conduct a landscape analysis but the results of the local search are close to methods known to avoid local optima. This corroborates our hypothesis, that the search space is regular and well-behaved, as we could not find any evidence that iterative improvement remains trapped in local optima, that perform significantly worse than the best know solutions.

The design methods that operate on behavior trees do not improve their solution quality significantly when the budget is increased. This could be an indication that all considered design methods for the restricted behavior tree architecture find solutions close to the actual optimum for that restricted architecture. This reasoning seems to be supported by the fact, that neither GP nor Maple, two design methods based on optimization algorithms that are known to avoid local optima, find any solutions outperforming the iterative improvement based design methods. Unfortunately, the design methods that operate on the restricted behavior tree architecture fail to achieve the same performance as those that operate on finite-state machines. This can be attributed to the smaller search space, defined through the restricted architecture (*Kuckling et al., 2018b*). While the design process converges more quickly towards the suspected optimum, it cannot reach the same solution quality as for the finite-state machines. However, we did not analyze how thorough the design methods explore this search space. This is mainly due to several limitations that prevent a straight-forward analysis, and therefore make this analysis to be out of the scope of this paper. Firstly, it is difficult to quantify how well the search space is explored. For finite-state machines, it is possible a finite-state machine can be represented through multiple orderings of the states. As no considered optimization algorithm accounts for this kind of equivalency, leading to the argument, that each of these equal representations would need to be counted independently. Yet, discovering one of these representations yields exactly the same performance as discovering any other or even all of the equal representations. This leads to the argument, that equivalent representations should only account for a single unit of the search space. Secondly, measuring the similarity between two instances of control software could influence the exploration metric of the search space. Two instances of control software that differ only minimally in exactly one parameter of the condition will most likely show very similar performance, although they are different instances of control software. Again, deciding whether or not these kinds of similarities should be accounted for when determining how well the search space is explored, is a non-trivial decision. Lastly, even if we decided on a method for accounting for similar and equivalent instances of control software in the search space, a simple analysis of the exploration of the search space is meaningless without analysis of the density and location of sufficiently good solutions in the search space. In fact, many regions of the search space are performing poorly with respect to any individual mission. Indeed, even the well-performing instances of control software of one mission are expected to perform poorly when applied to another mission. As such a method, exploring 10% of the search space, albeit in the low performing regions, will be

less desirable than a method exploring only 1% of the search space, but in the regions that contain the well-performing instances of control software.

For the design methods operating on finite-state machines, `Chocolate` is outperformed by the design methods based on iterative improvement for AAC, FORAGING, and GUIDED SHELTER for budgets of 25 k and 50 k. A reason for this difference in performance may be that the missions are creating a search space for the iterative improvement that is easily exploitable by the algorithm. That is, there seems to be only one way to successfully perform in the missions and the objective function does not contain local optima. This can be seen in the comparison of the two shelter mission SCA and GUIDED SHELTER. In SCA, the robots can only make use of a few indirect cues to determine the position of the shelter. Namely, two spots at the side of the shelter and a light in front of it. The main idea behind these cues is to allow the robots to determine the area in front of and behind the shelter. Yet, a successful shelter finding cannot solely rely on the ambient cues, as it is possible to miss the shelter when doing anti-phototaxis or to miss the black spots when passing from the back of the arena to the front (or starting in the front from the beginning). The mission GUIDED SHELTER, on the other hand, provides a more reliable way to reach the shelter. The white corridor leading to the shelter can be followed more or less exactly using the ANTI-PHOTOTAXIS behavior and even more so as the robot can detect if it leaves the corridor via the transition to grey floor, allowing the design process to create a contingency behavior for this case. This difference in the difficulty of the task shows also in the performance of the design methods using iterative improvement. In the mission GUIDED SHELTER, the design methods are able to generate near-optimal solutions for a budget of only 12,500 simulations and higher budgets did not lead to any noticeable improvement. In the mission SCA, on the other hand, all design methods (except `EvoStick`) show no improvement with higher budgets either. Visual inspection of the generated behaviors shows, however, that the designed control software fails to generated appropriate swarm level behaviors. While some robots end up in the shelter, the majority of them are distributed throughout the arena. This might be caused by the reduced amount of cues and additionally by the fact that the shelter can only house a fraction of the swarm.

The investigation of the generated control software and the resulting behaviors has not shown any indication, as to why finite-state machines suffer more strongly from the pseudo-reality gap than the generated behavior trees. We can speculate about possible reasons, but future work will be required to confirm or reject these hypotheses. One hypothesis would be that the bias/variance tradeoff can also be observed through the restrictions applied to the behavior tree architecture. In the restricted behavior tree architecture, the execution of the current behavioral module can only be terminated by meeting a single condition. In the generated finite-state machines, many states have however at least two different outgoing transitions with different associated conditions. Therefore, when assessed in pseudo-reality, there are at least two different transitions, which could trigger prematurely or delay triggering, thus altering the behavior of a robot.

Across all missions, all design methods based on iterative improvement perform similarly. Differences that may be observed for a given mission or a given budget do not generalize

to all observed results. This indicates that, for the considered budgets and missions, the starting solution does not seem to play an important role in the solution quality.

## CONCLUSIONS

In this paper, we explored the adoption of iterative improvement in the automatic modular design of control software for robot swarms. We have defined a neighborhood for two architectures: finite-state machines and the restricted behavior trees. We have shown that, within this neighborhood, every valid instance of control software can be transformed into any other one. Furthermore, we conceived several design methods based on an iterative improvement algorithm based on this neighborhood. The methods differ in the choice of the initial candidate solution, which can be chosen either as a minimal valid instance of control software, a randomly generated one or as the result of a previous design. The results show that the design methods based on iterative improvement perform similarly, indicating that the initial solution of the iterative improvement algorithm does not play a significant role in the missions considered. Our results do not indicate that any of the three considered choices for the initial solution candidate is better than the others. Furthermore, the performance of the design methods based on Iterated F-race and the performance of those based on iterative improvement are similar. This indicates that at least for the missions we considered, the search landscape is sufficiently smooth and early convergence to a local optimum is not an issue. However, for a budget of 50,000 simulation runs, `Hybrid-FSM` and `Chocolate` are outperformed by `Random-FSM` and `Minimal-FSM`. This indicates that, for larger design budgets, iterative improvement performs better than Iterated F-race. Additionally, the results highlight the shortcomings of the restricted behavior tree architecture. We showed that the restricted architecture is sufficient to perform adequately in all four missions at lower budgets. At higher budgets, however, the design methods operating on finite-state machines were able to exploit features that could not be represented in the restricted behavior tree architecture. Thus their performance exceeded that of the design methods operating on the restricted behavior tree architecture.

In future work, we will loosen the restrictions on the architecture of behavior trees. This will hopefully allow us to reduce the performance difference we observed between finite-state machines and behavior trees throughout most missions. Furthermore, we will consider optimizing behavior trees using grammatical evolution *O'Neill & Ryan (2003)* and reinforcement learning *Kaelbling, Littman & Moore (1996)*. These optimization algorithms have been recently applied to behavior trees and shown to provide promising results in different domains. Additionally, we will investigate other local search techniques, such as simulated annealing *Kirkpatrick, Gelatt & Vecchi (1983)* and iterated local search *Loureno, Martin & Stützle (2003)*, as these techniques have shown state-of-the-art performance on several optimization problems. We, therefore, expect similar high performance in the automatic modular design of control software for robot swarms.

## APPENDIX

### Finite-state machines

#### Completeness of the perturbation operators

In this section, we prove that any valid finite-state machine can be transformed into any other valid finite-state machine through the application of the perturbation operators P1–P11.

*Corollary FSM.1*

Given a valid finite-state machine, according to the criteria of this section, if a state has the maximum number of outgoing transitions, then at least two of its outgoing transitions point to the same end state.

*Proof.* In this setting, a valid finite-state machine has at most four states (see the definitions of validity further up). If a state has the maximum number of four outgoing transitions and no transition is allowed to be self-referencing, then each outgoing transition can point to one of the other states, which are at most three possibilities. The pigeonhole principle now states that there needs to be at least two of the four outgoing transitions pointing towards the same end state.

*Completeness of perturbation operators*

Let *FSM* and *FSM′* be two finite-state machines that are valid instances of control software according to the previous definition. *FSM* can be transformed into *FSM′* through the use of the perturbation operators defined in this section.

*Proof.* The proof of the aforementioned statement will be divided in several steps. After all steps have been executed, the initial finite-state machine *FSM* has been transformed into *FSM′*. The transformation steps are:

1. add states, if necessary;
2. transform into clique;
3. remove unneeded states, if necessary;
4. remove unneeded transitions, if necessary;
5. add transitions;
6. move initial state;
7. match modules;
8. match parameters.

*Step 1*

If *FSM* has fewer states than *FSM′*: repeatedly apply the operator P3 (add state) to add states to *FSM* until it has the same number of states as *FSM′*. This also adds one incoming and one outgoing transition to each state. If a state needs to be added, but all other states already have the maximum number of outgoing transitions, select one state $s$ and one transition $t$ outgoing of $s$ into $s'$ in such a way that $s$ has another transition to $s'$ (this is possible because of Corollary FSM.1). Remove this transition $t$ with the operator P2

(remove transition). Now add the additional state $s'$ using the operator P3 (add state), creating a transition from $s$ to $s'$ and a transition from $s$ to any other state in the finite-state machine.

*Step 2*

For each ordered pair of states $s, s'$ in the finite-state machine, add a transition from $s$ to $s'$ through the application of the operator P1 (add transition), if it does not already exist. If $s$ already has the maximum number of outgoing transitions, then P1 (add transition) is not applicable. Instead, select one transition from $s$ to $s'$, where $s'$ is another state such that at least two transitions point from $s$ to $s'$ (possible because of Corollary FSM.1) and change its endpoint to $s'$ through the application of P6 (move transition end). This step transforms the state transition graph of *FSM* into a directed clique.

*Step 3*

For every state, define a matching from the states of *FSM'* to the states of *FSM*. As Step 1 guarantees that *FSM* has at least the same number of states as *FSM'* and Step 2 did not alter the number of states in *FSM*, every state of *FSM'* can be matched to a state in *FSM*. Excess states in *FSM* are not matched and are marked for deletion. Using operator P4 (remove state), remove every such state $s$ that has been marked for deletion. This is possible, as the state is definitely not an articulation vertex, as the remaining states and their transitions still form a directed clique.

*Step 4*

For every state $s$ in *FSM* and every outgoing transition of $s$, if *FSM'* has at least one transition from $s = start(t)$ to $end(t)$, then match $t$ to one of these transitions. Else remove this transition through the application of the perturbation operator P2 (remove transition). This is possible, as this can never remove the last outgoing transition of $s$. Indeed the corresponding state in *FSM'* has at least one outgoing transition, or it is the only state in both finite-state machines *FSM* and *FSM'* so neither finite-state machine has a transition to remove anyway. Also, it cannot remove the only incoming transition into another state, as this transformation preserves at least one of the incoming transitions from $s$ to $s'$.
If this matching would delete all incoming transitions into $s'$, this would mean that there is no state $s'$ in *FSM'* either that has a transition to $s'$, thus creating an invalid configuration.

*Step 5*

For every state $s$ in *FSM*, if $s$ has fewer outgoing transitions than its corresponding state in *FSM'*, apply operator P1 (add transition) repeatedly, adding outgoing transitions to $s$ until it has the same number of outgoing transitions as its corresponding state. Match this newly generated transition to a transition in *FSM'* that has no matching yet. Use perturbation operator P6 (move transition end) to move the end state of the newly created transition to match the end state of the corresponding transition in *FSM'*. This again is possible because the newly created transition could not have been the only incoming transition into its original end state, otherwise this state would have been in an invalid configuration after Step 4.

*Step 6*

Use the perturbation operator P7 (change initial state) to move the initial state to the state *s* whose corresponding state in *FSM′* is the initial state of *FSM′*.

*Step 7*

For each state *s* in *FSM*, use perturbation operator P9 (change behavior) to match the behavior associated with *s* to the same behavior that is associated with its corresponding state in *FSM′*.

For each transition *t* in *FSM*, use perturbation operator P8 (change condition) to match the condition associated with *t* to the same condition that is associated with its corresponding transition in *FSM′*.

*Step 8*

For each state *s* in *FSM*, use perturbation operator P11 (change behavior parameters) to match the behavioral parameters of the behavior associated with *s* to the same parameters as of the behavior that is associated with the corresponding state of *s* in *FSM′*.

For each transition *t* in *FSM*, use perturbation operator P10 (change condition parameters) to match the conditional parameters of the condition associated with *t* to the same parameters as of the condition that is associated with the corresponding transition of *t* in *FSM′*.

## Behavior trees
### Completeness of perturbation operators

Let *BT* and *BT′* be two behavior trees that are valid instances of control software according to the previous definition. *BT* can be transformed into *BT′* through the use of the perturbation operators P1–P7.

*Proof.* The proof of the aforementioned statement will be divided into several steps. After all steps have been executed, the initial behavior tree *BT* has been transformed into *BT′*.

By definition, *BT* and *BT′* already contain the same single root (V1) and the same single top-level node that is, by definition, a sequence* (→*) node (V2). The following steps will ensure that all other parts of the behavior tree will be identical as well:

1. add selector subtrees, if necessary;
2. remove selector subtrees, if necessary;
3. update modules;
4. update parameters.

*Step 1*

If *BT* has fewer selector subtrees than *BT′*, then apply the perturbation operator P1 (add subtree) repeatedly until the number of selector subtrees is the same in the two trees.

*Step 2*

If $BT$ has more selector subtrees than $BT'$, then apply the perturbation operator P2 (remove subtree) repeatedly until the number of selector subtrees is the same in the two trees. Step 1 and Step 2 ensure that the two trees are structurally the same and all inner nodes already have the correct control node type associated. Additionally, create a matching between each leaf node in $BT$ to a leaf node in $BT'$, matching the two positionally identical leaf nodes in each behavior tree with each other.

*Step 3*

Traverse the leaf nodes of $BT$. If the current leaf node is a condition node, apply perturbation operator P4 (change condition) changing the condition to the one in the corresponding condition node in $BT'$. If the leaf node is an action node, apply perturbation operator P5 (change behavior) to change the behavior associated with that action node to the one associated with the corresponding node in $BT'$.

*Step 4*

Traverse the leaf nodes of $BT$. If the current leaf node is a condition node, apply the perturbation operator P6 (change condition parameter) onto that node matching the parameters of the corresponding node in $BT'$. If the leaf node is an action node, apply the perturbation operator P7 (change behavior parameter) onto that node to match the parameters of the corresponding node in $BT'$.

### Funding

The project has received funding from the European Research Council (ERC) under the European Union's Horizon 2020 research and innovation programme (grant agreement No. 681872). Jonas Kuckling, Thomas Stützle, and Mauro Birattari received support from the Belgian Fonds de la Recherche Scientifique – FNRS. The funders had no role in study design, data collection and analysis, decision to publish, or preparation of the manuscript.

### Grant Disclosures

The following grant information was disclosed by the authors:
European Research Council (ERC): 681872.
Belgian Fonds de la Recherche Scientifique – FNRS.

### Competing Interests

Thomas Stützle and Mauro Birattari are Academic Editors for PeerJ Computer Science.

### Author Contributions

- Jonas Kuckling conceived and designed the experiments, performed the experiments, analyzed the data, performed the computation work, prepared figures and/or tables, authored or reviewed drafts of the paper, and approved the final draft.

- Thomas Stützle conceived and designed the experiments, authored or reviewed drafts of the paper, and approved the final draft.
- Mauro Birattari conceived and designed the experiments, authored or reviewed drafts of the paper, and approved the final draft.

## Data Availability

Data and code are available in the Supplemental Files and at IRIDIA: http://iridia.ulb.ac.be/supp/IridiaSupp2020-007/index.html.

## Supplemental Information

Supplemental information for this article can be found online at http://dx.doi.org/10.7717/peerj-cs.322#supplemental-information.

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
