# Peer review of "Iterative improvement in the automatic modular design of robot swarms"

_PeerJ Computer Science, doi:10.7717/peerj-cs.322_

## Round 0.1 · original submission · Major Revisions

Both reviewers have found potential and relevant positive elements in your research and manuscript. Notwithstanding, both reviewers have addressed several issues in which the work can still be improved.

·

Basic reporting

The introduction provides useful background on the automatic design of collective behaviours for robot swarms, and sets the research in the context of previous work on AutoMoDe, clearly stating the aims of the article.

However, very few papers written by people outside of this research group are surveyed, which seems a little biased. Admittedly there is not much literature that is directly relevant to this work, but Jones et al. (2016) should at least be discussed in detail.

In some places there are long lists of citations such as:

"A general methodology for the manual design of robot swarms is still missing and existing approaches either operate under restrictive assumptions (Hamann and Worn, 2008; Kazadi, 2009; Berman et al., 2011; Brambilla et al., 2014; Reina et al., 2015; Lopes et al., 2016) or are labor intensive, time consuming, error prone, and nonreproducible (Brambilla et al., 2013; Francesca and Birattari, 2016; Bozhinoski and Birattari, 2018)"

"Neuro-evolutionary swarm robotics has been successfully applied to several missions (Quinn et al., 2003; Christensen and Dorigo, 2006; Hauert et al., 2009; Trianni and Nolfi, 2009), yet a number of limitations remain"

Please briefly explain the specific relevance of each of these citations, rather than listing them without further comment.

The rest of the article is well-written, and presents the research in a logical progression. The description of iterative improvement and the specific variant of stochastic hill-climbing used is particularly good. I'm not convinced that the proofs of completeness for the perturbation operators add much to the paper though - perhaps illustrating them with worked examples would improve their contribution.

The supplementary material is a useful addition, but videos of the generated behaviours would help significantly to appreciate the qualitative differences in the solutions produced by each algorithm.

The caption for Figure 1 should be more descriptive - please state here what each symbol represents (behaviours, conditions, selector, and sequence nodes).

The caption for Figure 2 should also describe the images in more detail - e.g. that these are ARGoS simulations with 20 e-puck robots, with light sources and coloured floor patches.

Figure 3 seems to have a formatting issue with the chart labels, which makes it look like BTFSM is one label, rather than BT FSM next to each other for Minimal/Random/Hybrid. This is fixed in Figure 4 and 5, so please update Figure 3 to match these.

Figure 3/4/5 - specifying "120sec" in each panel seems redundant, as this is already explained on line 477. Either remove this or factor it out to the figure captions.

The definition of the perturbation operators is a little inconsistent - some refer to a state 's' or a transition 't', while others are less formalised. Also, it is not clear how certain transitions/states are selected by the search algorithm. For example "Remove a transition from the finite-state machine: Remove a transition t. Do not remove the only outgoing or incoming transition of a state." - how is the transition 't' chosen? Sometimes the word "arbitrary" is used when referring to them - does this mean they are chosen at random (uniform distribution)?

The article suffers from not being sufficiently self-contained, and would therefore be difficult to understand for someone not familiar with previous AutoMoDe publications. The set of behaviours and conditions used to construct the controllers is not defined (behaviours like anti-phototaxis are only mentioned in the Discussion section), the details of the pseudo-reality simulation are vague, and information required for reproducibility such as the dimensions of the arena and floor patches is missing. While I appreciate that these details have been published in previous papers, please reiterate the key details here.

There are a few grammatical errors and typos throughout - please proofread carefully before final submission.

Experimental design

The experimental design is quite strong - the swarm is sufficiently large, standard tools (e-puck robots and ARGoS3 simulator) are used, appropriate case study swarm tasks were chosen, and the performance of iterative improvement is compared against that of AutoMoDe-Chocolate/Maple, GP, and EvoStick.

The data gathering is rigorous - experiments are carried out in both simulation and "pseudo-reality" in lieu of real hardware. However, there is no statistical analysis of the data obtained. The methods are not quite described in sufficient detail to be replicated, but the source code is available online in a GitHub repository.

Although the installation instructions were comprehensive, there was no clear documentation on how to reproduce the experiments presented in the paper. There is some documentation in each separate code repository, but instructions/scripts for reproducing the data presented in this article would be appreciated.

Validity of the findings

The discussion of the results tends to focus on the difference in performance between the FSM and BT variants of each algorithm, and their ability to cross the pseudo-reality gap, rather than the differences in performance between each algorithm.

Although the BT variants seem to produce lower-scoring behaviours, the spread of the data is much smaller and the results are therefore more consistent. Conversely, the FSM variants tend to produce a wide spread of data, suggesting that they do indeed get stuck in local optima. FSMs also seem to suffer worse than BTs when attempting to cross the pseudo-reality gap - why is this?

The article would benefit from deeper analysis of the experimental results, that provides valuable insight into the data. There is some relatively superficial analysis and conjecture, but not much in the way of concrete detail that explains the observed results.

It would be helpful to see examples of the generated FSMs and BTs, with an explanation of how these relate to the scenarios, and why they achieve good/poor performance (particularly in comparison to the other methods such as AutoMoDe-Chocolate/Maple).

There are a few conclusions that are not convincingly supported by empirical evidence:

"We found no indication that the search space is formed in such a way that it would hinder a local search algorithm to effectively find an optimum"

"We did not conduct a landscape analysis but the results of local search are close to methods known to avoid local optima. This corroborates our hypothesis, that the search space is regular and well-behaved, as we could not find any evidence that iterative improvement remains trapped in local optima."

"The design methods that operate on behavior trees, do not improve their solution quality significantly, when the budget is increased. This indicates that all considered design methods for the restricted behavior tree architecture find solutions close to the actual optimum for that restricted architecture."

Without knowing the global optima for each scenario, not being able to improve upon the results with larger simulation budgets is not sufficient evidence to suggest that the algorithms are not getting stuck in local optima. Particularly as there is no comparison to a global search algorithm, or something like a genetic algorithm that includes elements of global search (crossover) as well as local search (mutation), it seems difficult to argue that your local search algorithms are not all getting stuck in local optima.

Can you quantify the size of the search space, and comment on how thoroughly this is searched by the algorithms analysed?

Additional comments

This research is an interesting incremental extension to the existing body of AutoMoDe research, and deeper analysis of the experimental results could provide some valuable insights that would inform future work.

·

Basic reporting

none

Experimental design

none

Validity of the findings

none

Additional comments

This paper describes a comparative study for the iterative improvement optimisation technique for automatic design method of two control architectures namely Finite-State Machines FSM and Behavior Trees BT for swarm robotics systems. The study evaluates the performance of a set of optimisation algorithms (i.e., Chocolate, Maple, Minimal-FSM, Minimal-BT,
Random-FSM, Random-BT, Hybrid-FSM, Hybrid-BT, GP and EvoStick) for the two control architectures (i.e., FSM, BT). The study describes the properties of FSM and BT which must have in order to considered a valid instance of control software. Additionally, the study explains a set of possible perturbation operators performed on FSM and BT in order to transform them to another valid finite-state machine and all the optimisation algorithms are introduced briefly.

Generally, this paper is properly organised and well written. A set of extensive experiments is described to provide sufficient conclusive evidence on the outcomes of this research. Supplementary material provides the necessary resources which are sufficient to replicates the work described in the paper. In my opinion, the results comprise useful message to the research community in the field of automatic design methods to generate collective behaviour of robot swarm.

However, I have few comments to be addressed in the followings:

1- It has been mentioned in table1 (p.8), the period of control cycle is 100 ms. However, in mission section (p. 13- L. 477) you referenced table1 and you mention the experiment runtime is 120 s. What do you mean by the former number? And why the two numbers are different?

2- I do not understand the use of 2 black spots in the SCA task. Despite the explanation in the discussion section which indicates the spots are not a very useful cue to find the shelter. It is also not clear how the robot develops a strategy to search for the shelter and hide inside it. Please, elaborate on this.

3- It is not described in the foraging task what is the mechanism used to evaluate the performance of the foraging mission. In particular, how to measure that the robot visit (moved over) foraging site (one of the black spots) then visit the nest site (moved white area) subsequently. Please, explain this.

4- It would be good if you include video of successful trials in the supplementary material. This could help for visual inspection of the most effective developed strategies.

5. In fig 3, 4, and 5. It would be good if the y-axis (score) is normalised between [0,1]. I understand the difference in the score range of values is due to the difference in the objective function of each mission. However, the current y-axis is confusing readers when comparing the graphs in each figure. The overall comparison of the missions in one figure indicates that one mission outperformed others when a unified score metic is assumed between the graphs.

6. I believe the inclusion of Evostick in the study is irreverent in the context of the paper since it uses ANN. The main objective of the study is to compare the iterative improvement optimisation for FSM and BT therefore the presence of ANN in one instance (in Evostick) is consistent with the theme of the paper.

---

## Round 0.2 · accepted · Accept

Previous issues addressed by reviewers have been now sorted out. I consider the paper valuable and interesting

·

Basic reporting

Previous concerns have been addressed.

Experimental design

Previous concerns have been addressed.

Validity of the findings

Previous concerns have been addressed.

Additional comments

Thank you for taking the time to address my concerns - the revisions made to the paper and the supplementary material are a significant improvement.

I still think it would be helpful to see examples of the generated FSMs and BTs, even if only in the supplementary material. In your rebuttal later you said that "we felt it inappropriate to directly include figures of all discussed instances of control software. Instead we made them available on the supplementary materials page", however the closest I can find at the linked URL is text-based encodings of the controllers (within data.zip). Could a visualisation of the controller be included with each controller.txt, or could you provide a script that allows the reader to generate a visualisation from these text encodings?

My only other concern is that the newly-added Table 2 that lists "noise models" doesn't provide context for the values. Presumably these are the noise parameters used for the ARGoS sensors/actuators, but I don't think these currently have consistent semantics (see this open issue raised on the ARGoS GitHub repository: https://github.com/ilpincy/argos3/issues/120). Perhaps the DEMIURGE argos3-epuck plug-in resolves this issue for the e-puck-specific sensors/actuators, but it would help to state whether uniform or Gaussian noise distributions are used, and what the values in the table represent (e.g. standard deviations).